# Sulfoxaflor Applied via Drip Irrigation Effectively Controls Cotton Aphid (*Aphis gossypii* Glover)

**DOI:** 10.3390/insects10100345

**Published:** 2019-10-14

**Authors:** Hui Jiang, Hanxiang Wu, Jianjun Chen, Yongqing Tian, Zhixiang Zhang, Hanhong Xu

**Affiliations:** 1State Key Laboratory for Conservation and Utilization of Subtropical Agro-Bioresources, South China Agricultural University, Guangzhou 510642, China; jianghuizukai@stu.scau.edu.cn (H.J.); hanxiang.wu@scau.edu.cn (H.W.); tyq2006@scau.edu.cn (Y.T.); 2Key Laboratory of Natural Pesticide and Chemical Biology, South China Agricultural University, Guangzhou 510642, China; 3Department of Environmental Horticulture and Mid-Florida Research and Education Center, Institute of Food and Agricultural Sciences, University of Florida, Apopka, FL 32703, USA; jjchen@ufl.edu

**Keywords:** *Aphis gossypii* Glover, *Chrysoperla sinica* T., *Coccinella septempunctata* L., drip irrigation, natural enemy, sulfoxaflor

## Abstract

*Aphis gossypii* Glover is a major pest of cotton and can severely affect cotton yield and lint quality. In this study, the efficacy of sulfoxaflor applied via drip irrigation and foliar spray on controlling cotton aphids was evaluated in 2016 and 2017 in Xinjiang, China. The distribution of sulfoxaflor in cotton roots, stems, leaves, and aphids, as well as its effects on two natural enemies of aphids, were also investigated. Results showed that sulfoxaflor applied through drip irrigation mainly concentrated in leaves and provided effective control of cotton aphids for 40 days, compared to 20 days when applied through foliar spray. Furthermore, drip application resulted in much lower sulfoxaflor concentrations in aphids than foliar spray. As a result, ladybird beetle and lacewing populations were higher in drip applied plants than in foliar sprayed plants. Additionally, the cost of drip irrigation was lower than foliar spray as cotton plants are commonly irrigated via drip irrigation in Xinjiang. Our results showed that application of sulfoxaflor through drip irrigation is an effective way of controlling cotton aphids in Xinjiang due to a prolonged control period, safety to two natural enemies, and lower cost of application.

## 1. Introduction

Cotton is an important fiber-producing plant worldwide. Cotton aphid (*Aphis gossypii* Glover) is one of the key pests of cotton, particularly at the seedling stage [1]. The aphid damages leaves, spreads virus, and has a serious impact on crop yield and lint quality [2,3,4,5]. Xinjiang Uyghur Autonomous Region (equivalent to a province) is the world’s most important cotton-producing region, accounting for more than 50% of the overall yield in China and 10% of the annual global cotton lint production [6,7]. However, the chronic application of pesticides has resulted in cotton aphid resistance to common pesticides, and cotton aphid is becoming a major hindrance to cotton production.

Current control of the aphid relies largely on chemical application. Insecticides, mainly neonicotinoids are applied through tractor-mounted sprayers in Xinjiang. Foliar application through tractor-mounted sprayers has been challenging due to the high plant density in the field, vast production area, non-target insecticide exposure, and labor intensity [8]. Tractor-mounted sprayers can also significantly damage a number of cotton plants. Additionally, due to the high fecundity of cotton aphids [9] and the short residual effects of sprayed insecticides [10], cotton farmers need to spray multiple times to control the aphid. Therefore, it is important to identify alternative methods for application of insecticides in this region.

Drip irrigation of insecticides represents an alternative method for integrated pest management because it is relatively safe and causes less pollution [11]. Currently, more than 90% of cotton fields in Xinjiang are irrigated through a drip system [12,13]. Moreover, several insecticides, such as neonicotinoids and anthranilic diamides were successfully applied via drip irrigation in pest control [14,15,16]. Thus, it could be feasible to deliver insecticides to cotton through existing drip irrigation systems.

Sulfoxaflor is a novel systemic insecticide with nicotinic acetylcholine receptor-modulating activity and has high activity against a wide range of sap-feeding insects, such as aphids [17,18,19,20]. Sulfoxaflor has been widely used to control different aphids due to its high potency and lack of insecticidal cross-resistance [19,21]. However, the effect of sulfoxaflor against cotton aphid, the distribution of sulfoxaflor in cotton plants, and its effects on natural enemies of aphids have not been studied in the field, especially in the context of application through drip irrigation.

This study was intended to evaluate the efficacy of sulfoxaflor applied via drip irrigation in relation to foliar spray for control of cotton aphid, the distributions of sulfoxaflor in cotton roots, stems, leaves, and aphids, as well as the effects of the two application methods on two natural enemies of aphid: ladybird beetles (*Coccinella septempunctata* L.) and lacewing (*Chrysoperla sinica* T.).

## 2. Materials and Methods

### 2.1. Cotton Seeds and Insecticides

Transgenic Bt cotton seeds of variety Xinliuzao-42 were supplied by the Cotton Research Institute of Xinjiang Academy of Agricultural Reclamation Sciences and Xinjiang Huiyuan Agricultural Science and Technology Development Co., Ltd. Sulfoxaflor of certified reference standard (purity, 98%, CAS number 946578-00-3) was purchased from Shanghai Mingbo Biotechnology Co., Ltd. Sulfoxaflor 50% water dispersible granule (WDG) was provided by Dow AgroSciences (Zionsville, IN, USA).

### 2.2. Field Experimental Site

The experiments were carried out in 2016 and 2017 in Bole County, Xinjiang, northwest China (44°02′~45°23′ N and 79°53′~83°53′ E). The cotton field has a loamy soil containing 81.79 mg.kg^−1^ alkali-hydrolyzable nitrogen, 11.64 mg.kg^−1^ available phosphorus, 293.15 mg.kg^−1^ available potassium, and 14.03 g.kg^−1^ soil organic matter. The soil pH is 8.49. On 5 April 2016 and on 24 April 2017, cotton seeds were sown in the field installed with a drip irrigation system. The drip irrigation tape employed a mode of “one film-three pipe-six rows”. The film was 2 m in length and the pipe spacing was 66 cm. Each tape was in the middle of two rows of cotton plants with a spacing of 10 cm. The drip emitter was spaced 20 cm apart and the flow rate of each drip emitter was 2.0 L.h^−1^. The cotton plants were irrigated and fertilized in accordance with the local cotton practices.

### 2.3. Sulfoxaflor Application

The experiments were initiated on 3 June 2016 and 21 June 2017, respectively, when the cotton plants were at principle growth stage 3, i.e., main stem elongation and the foliage of 30% of the plants meet between rows [22], and the incidence of cotton plants with aphids was over 20%. The experiment consisted of four treatments: (1) 1400 g.ha^−1^ (700 g a.i. ha^−1^) sulfoxaflor (Closer 50% WDG) applied via drip irrigation; (2) 150 g.ha^−1^ (75 g a.i. ha^−1^) sulfoxaflor (Closer 50% WDG) applied via foliar spray; (3) drip irrigation of water as a control; (4) foliar spray of water as another control. The four treatments were replicated three times in 12 plots in a completely randomized design. Each plot was 200 m^2^ (3200–3500 plants per 200 m^2^) and separated by a 4 m buffer zone. For the drip irrigation treatments, 2500 L sulfoxaflor solution was applied first to each plot followed by the application of 1000 L water through drip irrigation. The control had the same irrigation regime without any chemicals. For the foliar spray treatments, each plot was sprayed with 13 L (650.3 L ha^−1^) sulfoxaflor solution using a knapsack electric sprayer (single nozzle with 0.15–0.4 Mpa, 4–5 m width). The control had the same application regime without any chemicals. The schedule of sulfoxaflor treatments is presented in Table 1.

The application rate of sulfoxaflor via drip irrigation was higher (700 g a.i. ha^−1^) than that of foliar spray. This rate was based on our preliminary study data and also in reference to other reports [23,24,25] that soil could adsorb insecticides and affect their availability. The soil used in this study is saline and alkaline, and its pH is higher than 8. Lalah et al. reported that pH was one of the important factors affecting the activity of pesticides [26]. Timmeren et al. reported the same trend that a single application rate of insecticide via soil treatment needed to be higher than that of foliar spray [27].

#### 2.3.1. Investigations of Cotton Aphid and Natural Enemies

Leaves of cotton plants were divided into three groups: (1) lower leaves: the leaves of the first and second branch from the base of stem; (2) middle leaves: the leaves of the fifth and sixth branch; (3) top leaves: recently expanded and growing leaves. The investigation method of aphids on cotton leaves referred to the guidelines of field efficacy trials (II) part 75 [28]. To quantify aphids, 40 cotton plants were randomly chosen from each plot, and the number of aphids at 0, 1, 3, 5, 7, 15, 20, 30 and 40 days after treatments were counted. Meanwhile, natural enemies (adult ladybird beetles, ladybird beetle larvae, adult lacewing, and lacewing larvae) were also monitored. The investigation method of natural enemies referred to the five observation points. Five observation points were randomly selected in each plot, and each point included 40 cotton plants.

#### 2.3.2. Sample Collection of Cotton Plants, Seeds, and Aphids

**Cotton plants**: Ten cotton plants were randomly selected per plot at 12 h and day 1, 3, 5, 7, 15, 30, 40, and at harvest after drip irrigation of sulfoxaflor, and at 2 h and day 1, 3, 5, 7, 15, 20, 30, and at harvest after foliar spray of sulfoxaflor. Samples of leaves from the top, middle, and lower groups, stems, and roots, approximately 20 g each, were collected and placed in self-sealing polyethylene bags and frozen immediately.

**Aphids**: More than 2.00 g of cotton aphids were collected from 120 cotton plants per plot of the four treatments at 0.5 h and 7 days after sulfoxaflor application in 2016 and 2017, placed in a centrifuge tube (10 mL), and frozen immediately.

**Cotton seeds**: Cotton seeds (50 g) in each plot were collected and placed in self-sealing polyethylene bags after they were machine-delinted, and they were then frozen immediately.

All collected samples were transported to the South China Agricultural University Pesticide Analytical Laboratory and placed in a −20 °C freezer until extraction.

### 2.4. Pretreatment of Cotton Plants, Seeds and Aphid Samples

**Cotton plants and seeds:** Samples of cotton root, stem, leaves, and seeds were placed in 50 mL conical centrifuge tubes with 5.0 mL water and 20.0 mL AR-grade acetonitrile. All tubes were sonicated for 30 min, 2.0 g sodium chloride was added, and they were vortex mixed for 2 min at 1000 rpm. All tubes were centrifuged for 5 min at 4000 rpm and acetonitrile extract (supernatant) was dried under vacuum. The sample was eluted twice with 2.0 mL of CG-grade acetonitrile; 1.0 mL supernatant was transferred into a 2.0 mL plastic centrifuge tube containing PSA (0.10 g) and C_18_ (0.10 g) and centrifuged for 5 min at 6000 rpm. Finally, 1 mL of the supernatant was passed through a 0.22 μm syringe filter (Nylon). Each sample had three replications.

**Aphids:** Samples of cotton aphids were placed in 10 mL conical centrifuge tubes with 0.5 mL water and 2.0 mL AR-grade acetonitrile. All tubes were sonicated for 30 min; 0.5 g sodium chloride was added and vortex mixed for 2 min at 1000 rpm. All tubes were centrifuged for 5 min at 4000 rpm and acetonitrile extract (supernatant) was dried by nitrogen evaporators. Then, the sample was eluted twice with 2.0 mL of CG-grade acetonitrile and 1.0 mL supernatant was transferred into a 2.0 mL plastic centrifuge tube containing PSA (0.10 g) and C_18_ (0.10 g), and centrifuged for 5 min at 6000 rpm. Finally, 1 mL of the supernatant was passed through a 0.22 μm syringe filter (Nylon) and dried by nitrogen evaporators into 0.5 mL. Each sample had three replications.

### 2.5. Sulfoxaflor Analysis

All samples were analyzed by Agilent 1100 Series high-performance liquid chromatography (HPLC) equipped with UV and PDA detectors. The XDB-18 reversed phase column (4.6 mm × 250 mm, 5 μm) with a temperature set at 30 °C was used for separation. The mobile phase was water/acetonitrile (85:15 by volume) and the flow rate was 1.0 mL.min^−1^. The injection volume was 10.0 μL and the wavelength of detection was 260 nm.

The extraction method and analytical method were validated. The correlation coefficient (*R*^2^) for the standard curve of sulfoxaflor ranging from 0.05 mg L^−1^ to 10.00 mg L^−1^ was 0.9974. The recovery experiments of sulfoxaflor in cotton roots, stems, leaves and seeds were conducted at the three spike levels of 5, 0.5, and 0.05 mg kg^−1^, and each level was replicated three times. These samples were processed as described above. Mean recovery values for sulfoxaflor were 80.00%–105.11% in cotton root, stem, leaf, seed and aphid samples, which were in the acceptable range (70%–120%) specified by the SANCO guidelines [29]. The limit of quantification (LOQ) of sulfoxaflor in roots, stems, leaves, aphids and seeds was 41.67 μg kg^−1^, 48.00 μg kg^−1^, 50.67 μg kg^−1^, 43.33 μg kg^−1^ and 45.03 μg kg^−1^, and the limit of detection (LOD) was 12.50 μg kg^−1^, 14.40 μg kg^−1^, 15.20 μg kg^−1^, 13.00 μg kg^−1^ and 13.51 μg kg^−1^, respectively.

### 2.6. Cost Assessment on the Two Application Methods

Based on the efficacy of the two application methods in aphid control, we assumed that the efficiency of double foliar application of sulfoxaflor was equivalent to that of single drip irrigation. We also investigated the cost of local labor involvement, local machinery and diesel, and local mechanical damage resultant from the economic loss of local farmers in Xinjiang. The cost for application of sulfoxaflor via drip irrigation and conventional tractor-mounted sprayers was then assessed.

### 2.7. Data Analyses

The corrected mortalities of aphids were calculated by the Abbott formula. All data represent means ± S.E. (n = 3). All statistical analyses were carried out using SPSS statistical software (version 15.0, SPSS Inc., Chicago, IL, USA). Data analysis was based on individual year. Statistically significant corrected mortality of cotton aphids, the number of natural enemies, and the concentration of sulfoxaflor in cotton plants and aphids were assessed by Tukey’s HSD test (*p* < 0.05).

## 3. Results

### 3.1. Control Efficacy of Sulfoxaflor with the Two Application Methods

Cotton aphid outbreaks occurred from mid-May to late-July in 2016 and from late-May to late-July in 2017 at this study site. Sulfoxaflor applied through drip irrigation at the rate of 700 g a.i. ha^−1^ in 2016 and 2017 effectively controlled cotton aphids for 40 days (Table 2). The corrected mortalities on the top and middle leaves were significantly higher than those on lower leaves except day 1, 20, and 30 in 2016 and day 1, 15, 20, and 30 in 2017 (*F* < 6.65, *p* > 0.061) (Table 2). The corrected mortalities on top leaves were not different from those on middle leaves for all days except day 3 in two years (2016: Day 3: *F* = 13.27, *p* = 0.022; 2017: Day 3: *F* = 9.74, *p* = 0.035) (Table 2). Meanwhile, sulfoxaflor applied via foliar spray at a rate of 75 g a.i. ha^−1^ effectively controlled cotton aphids for 20 days (Table 2). There was no significant difference in corrected mortalities among the top, middle, and lower leaves except day 1, 3, 5, and 7 in both years (Table 2). Based on the corrected mortalities, foliar spray provided more rapid control than drip irrigation. However, sulfoxaflor applied through drip irrigation resulted in higher corrected mortalities than those of foliar spray on all cotton leaves after day 7 and provided longer control efficacy than foliar spray (Table 2).

### 3.2. Distribution of Sulfoxaflor in Cotton Roots, Stems, and Leaves

Two-year data showed that the concentration of sulfoxaflor applied via drip irrigation in cotton roots, stems, and leaves increased from 12 h after application to a peak on day 5 or 7, and then gradually declined thereafter till day 40 (Figure 1a,b, Figure 2a,b). The concentration of sulfoxaflor in cotton leaves was significantly higher than in cotton roots and stems for all days except at 12 h and on day 1 based on two-year data (Figure 1a, Figure 2a) (*F* > 25.02, *p* < 0.002). Moreover, sulfoxaflor concentrations in top leaves were significantly higher than those in middle and lower leaves throughout the experiment except at 12 h in 2016 and on day 40 in 2017 (Figure 1b, Figure 2b) (*F* > 6.46, *p* < 0.033). Sulfoxaflor concentrations remained at 0.22 and 0.19 mg kg^−1^ in cotton leaves on day 40 in 2016 and 2017, respectively (Figure 1a, Figure 2a).

On the contrary, the concentration of sulfoxaflor applied via foliar spray in cotton stems and leaves declined gradually throughout the sampling periods in both 2016 and 2017. The highest values occurred in the first sampling time (2 h) (Figure 1c,d, Figure 2c,d). There was no significant difference between stems and leaves, but sulfoxaflor concentrations in leaves and stems were significantly higher than in roots (*F* > 33.12, *p* < 0.002) (Figure 1c, Figure 2c). This was because sulfoxaflor was directly sprayed on the surface of cotton stems and leaves, and the soil was covered by the film. By comparison, sulfoxaflor concentrations in top leaves were relatively higher than those in middle and lower leaves for all sample days except day 15 and day 20 in 2017 (Figure 1d, Figure 2d). After day 20, sulfoxaflor applied via foliar spray was not detected (Figure 1c, Figure 2c).

### 3.3. Effects of Sulfoxaflor on Ladybird Beetles and Lacewings

Ladybird beetles and lacewings were found on cotton plants in both 2016 and 2017. The two natural enemies of cotton aphid were more abundant in control treatment plots than in sulfoxaflor applied plots (Figure 3, Figure 4, Figure 5 and Figure 6). The predators in plants treated with sulfoxaflor through drip irrigation tended to have relatively lower abundance than those of the control treatment, but they were not statistically different (Figure 3, Figure 4, Figure 5 and Figure 6). On the contrary, the numbers of adult ladybird beetles on the cotton plants sprayed with sulfoxaflor were significantly lower than the control from day 3 to day 20 in 2016 and from day 1 to day 20 in 2017 (*F* > 6.37, *p* < 0.017) (Figure 3). The counts of ladybird beetle larvae on plants sprayed with sulfoxaflor were significantly lower than the control from day 1 to day 7 in 2016 and from day 1 to day 15 in 2017 (*F* > 8.92, *p* < 0.041) (Figure 4). Additionally, the numbers of adult lacewings on the cotton plants sprayed with sulfoxaflor were significantly lower than that of the control plants from day 1 to day 15 in 2016 and from day 1 to 7 in 2017 (*F* > 9.55, *p* < 0.038) (Figure 5). Similarly, the counts of lacewing larvae on the plants treated by foliar spray were significantly lower than the control from day 3 to day 15 in 2016 and from day 1 to day 20 in 2017 (*F* > 5.05, *p* < 0.031) (Figure 6).

### 3.4. Sulfoxaflor Concentration in Cotton Aphids

Sulfoxaflor concentrations in cotton aphids collected from plants 0.5 h after foliar spray of sulfoxaflor were significantly higher than those collected from drip applied plants in both 2016 and 2017 (*F* = 192.18, *p* < 0.001 in 2016, *F* = 97.63, *p* < 0.001 in 2017) (Figure 7). However, sulfoxaflor concentrations in aphids collected on day 7 from plants treated by the two application methods were not significantly different (Figure 7).

### 3.5. Sulfoxaflor Residue in Cotton Plants at Harvest

Sulfoxaflor was not detected in cotton roots, stems, leaves, and seeds at cotton harvest in 2016 and 2017, suggesting that sulfoxaflor applied at the application rate of 700 g a.i. ha^−1^ via drip irrigation or 75 g a.i. ha^−1^ via foliar spray was safe at harvest as the maximum residue limit of sulfoxaflor in cotton seed was set to be 0.40 mg.kg^−1^ [30]. Additionally, cotton yields in 2016 and 2017 resulted from drip application of sulfoxaflor were about 5% higher than those treated with foliar spray.

### 3.6. Cost of the Two Application Methods

In consideration of the control efficiency, mechanical damage to cotton plants, input of labor and machinery costs, as well as the price of cotton seeds harvested, we assessed the accumulative cost resulted from the application of sulfoxaflor through the two application methods. The results showed the accumulative cost for application of sulfoxaflor through foliar spray was higher than drip irrigation (Table 3).

## 4. Discussion

Results from both 2016 and 2017 showed that a single application of 700 g a.i. ha^−1^ sulfoxaflor via drip irrigation provided more effective control of cotton aphid than single foliar spray (75 g a.i. ha^−1^). Our results agreed with other reports [31,32,33,34] that drip application was more effective than foliar spray in pest management. Such an effective and prolonged control of aphid by drip application of sulfoxaflor could be attributed to the interaction of several factors: (1) sulfoxaflor is a novel systemic insecticide soluble in soil, so it can be readily absorbed by roots and transported to plant leaves [18,35]. (2) The mouthpart of the aphid pierce into xylem and feed on the sap [36]. Sulfoxaflor applied through drip application was transported through plant xylem, so the extraction of sap by aphid resulted in direct poison. (3) Drip applied sulfoxaflor was mainly concentrated in cotton leaves (Figure 1a, Figure 2a) (*F* > 25.02, *p* < 0.002), which is similar to the drip-applied dimethoate that was concentrated in leaves [37]. This is because top leaves or recently expanded and growing leaves have high transpiration rates, and the higher transpiration flow would bring more absorbed sulfoxaflor to these leaves. Meanwhile, cotton aphids have the propensity for feeding on growing leaves [38]. Thus, drip application of sulfoxaflor achieved an on-target control of aphid as is shown in Table 2, where the corrected mortalities on top leaves were higher than those of other leaves. (4) The sustainability of sulfoxaflor inside plant parts could minimize the problems associated with foliar spray, such as being washed away due to precipitation and degradation caused by direct sunlight and/or microbial activities [39]. Figure 1b and Figure 2b showed that sulfoxaflor concentrations in leaves of dripped plants were higher and sustained rather high concentrations for almost 40 days compared to those in plants treated with foliar spray. Our results concur with the reports of others [33,35,40,41] that drip irrigation offers effective and prolonged control of pests.

Environmental conditions affected sulfoxaflor concentrations in plant organs. Sulfoxaflor concentrations in cotton roots, stems, and leaves of drip applied plants in 2016 were relatively lower than those of 2017. On the contrary, sulfoxaflor concentrations in cotton roots, stems and leaves of foliar spray treatment in 2016 were relatively higher than those concentrations in 2017 (Figure 1 and Figure 2). These differences could be due to temperature effects because the daily temperatures were 3 °C higher in 2017 during the whole experimental period compared to 2016. For drip irrigation, the high daily temperature might result in higher transpiration rates, thus higher concentrations in cotton plants [42]. In foliar sprayed plants, the higher daily temperature might cause a rapid desiccation of spray insecticide droplets and high dissipation rates due to higher evaporation and also photodegradation [43,44].

This study also evaluated the effects of sulfoxaflor applied by two different methods on natural enemies of aphids. The results showed that plants treated with sulfoxaflor through drip irrigation had a higher population of ladybirds and lacewings than those treated by foliar spray. This result could be due to the following factors: (1) the natural enemies had no direct contact with sulfoxaflor with drip irrigation. On the contrary, foliar spray directly delivered sulfoxaflor onto the surface of aphids, cotton leaves, and natural enemies, which led to higher death rates of natural enemies. Our results showed higher levels of sulfoxaflor on cotton leaves at 2 h after foliar spray (Figure 1 and Figure 2). (2) Cotton aphids containing insecticides were ingested directly by predators as supplemental diet. The analysis results showed that foliar spray resulted in significantly higher sulfoxaflor concentrations in aphids compared to drip irrigation in 0.5 h (Figure 7). (3) The low prey density in the foliar spray plots failed to attract natural enemies [45]. The abundance of prey is the main factor that determines the distribution of natural enemies, and natural enemies tend to emigrate to, remain, and oviposit in areas with sufficient food resources [46,47]. Therefore, the significantly lower densities of natural enemies found in the foliar-treated plots were very likely to have resulted from low prey densities, high residue levels of sulfoxaflor in aphids, and high residue levels of sulfoxaflor in leaves.

Our results further showed that drip application of sulfoxaflor could reduce the frequency of insecticide sprays and result in lower labor, machinery, and diesel costs and less mechanical damage to cotton plants. Therefore, the accumulative cost of sulfoxaflor applied via drip irrigation saved $19.59 per hectare compared to foliar spray in Xinjiang (Table 3). Considering the fact that there are more than 1.7 million hectares of cotton fields in Xinjiang [48,49], the use of drip irrigation of sulfoxaflor could significantly reduce the production cost for aphid control. Moreover, the final residue of sulfoxaflor in cotton roots, stems, leaves and seeds was below the detection limit at harvest, which is below the maximum residual limit of sulfoxaflor in cotton seeds [30]. This indicated that drip application of 700 g a.i. ha^−1^ sulfoxaflor was safe at cotton harvest.

## 5. Conclusions

This is the first systematic evaluation of the efficacy, distribution, and effects of sulfoxaflor applied through drip irrigation and foliar spray in control of cotton aphids in an important cotton production region. Results showed that drip irrigation of sulfoxaflor could provide more effective and prolonged control of cotton aphids than foliar spray, have less detrimental effects on ladybirds and lacewings, two natural enemies of aphids, and can also reduce the production cost of cotton in Xinjiang. Drip irrigation of sulfoxaflor could represent a feasible, affordable and sustainable way of controlling cotton aphids in arid and semi-arid zones, such as Xinjiang.

## Figures and Tables

**Figure 1 insects-10-00345-f001:**
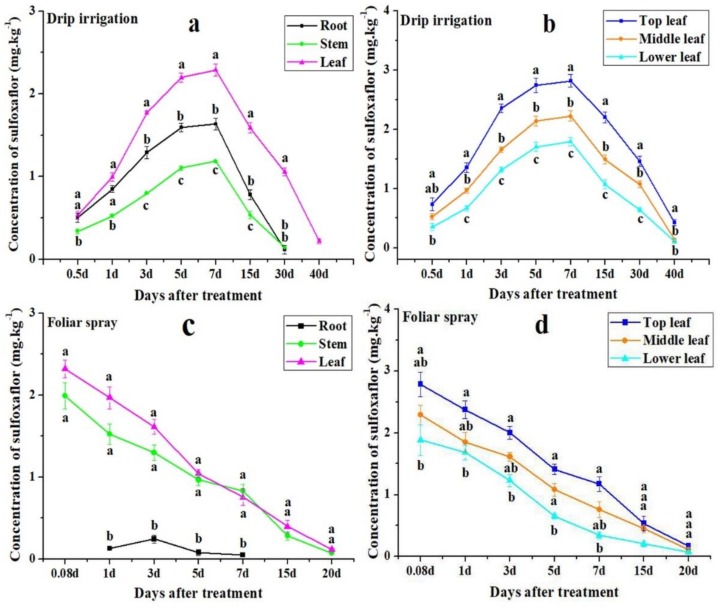
Distribution of sulfoxaflor applied through drip irrigation in roots, stems, and leaves (**a**) and in top, middle, and low leaves (**b**), and via foliar spray in roots, stems, and leaves (**c**) and in top, middle, and low leaves (**d**) in 2016. All data are expressed as mean ± S.E. (n = 3). Values within the same time period labeled with different letters are significantly different based on Tukey’s HSD test at *P* < 0.05. Note: concentrations of sulfoxaflor in leaves (**a**,**c**) were the average value of top, middle, and lower leaves (**b**,**d**). The same below.

**Figure 2 insects-10-00345-f002:**
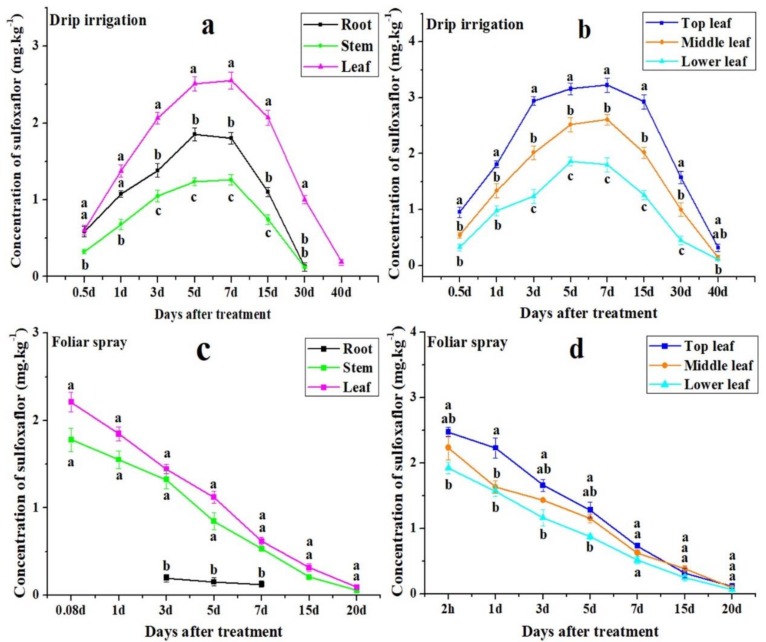
Distribution of sulfoxaflor applied through drip irrigation in roots, stems, and leaves (**a**) and in top, middle, and low leaves (**b**), and via foliar spray in roots, stems, and leaves (**c**), and in top, middle, and low leaves (**d**) in 2017. All data are expressed as mean ± S.E. (n = 3). Values within the same time period labeled with different letters are significantly different based on Tukey’s HSD test at *p* < 0.05.

**Figure 3 insects-10-00345-f003:**
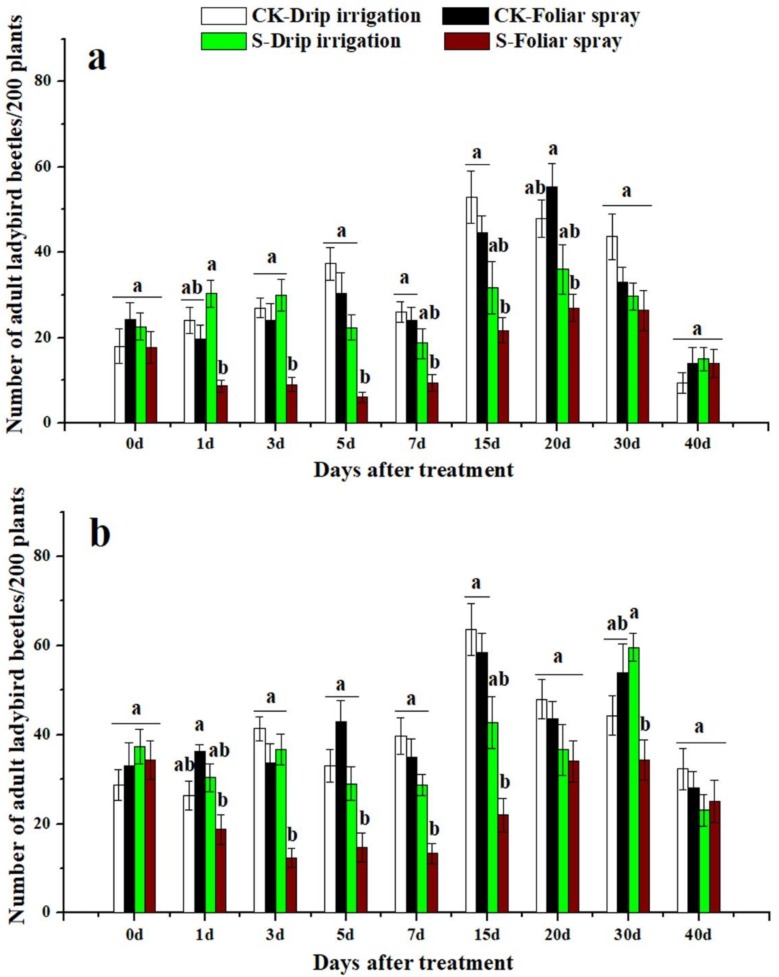
The numbers of adult ladybird beetles on cotton plants not treated with any chemical (CK-Drip irrigation and CK-Foliar spray) and treated with sulfoxaflor through drip irrigation (S-Drip irrigation) and foliar spray (S-Foliar spray) in 2016 (**a**) and 2017 (**b**). All data are expressed as mean ± S.E. (n = 3). Values within the same time period labeled with different letters are significantly different based on Tukey’s HSD test at *p* < 0.05.

**Figure 4 insects-10-00345-f004:**
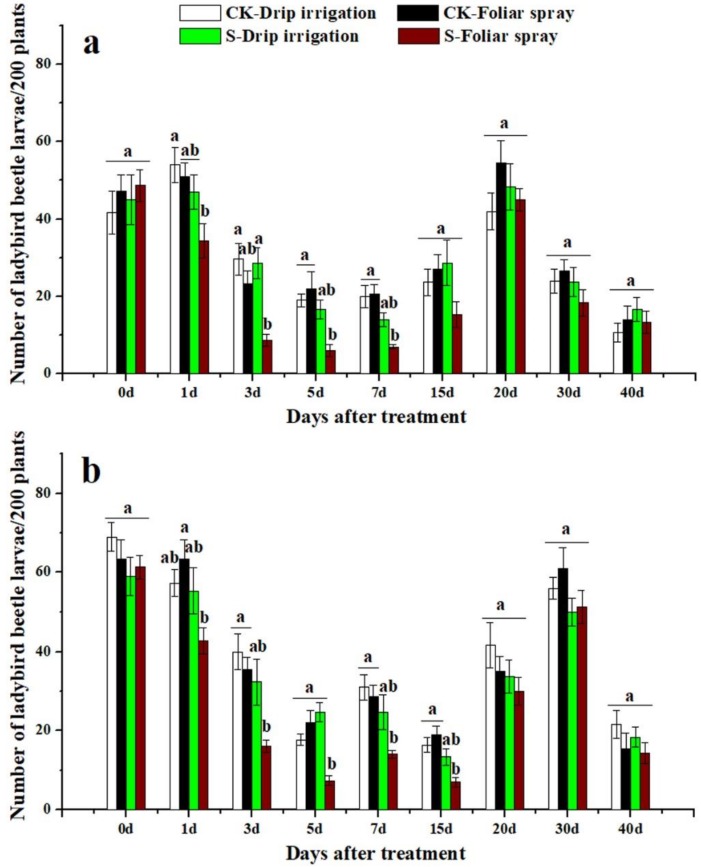
The numbers of ladybird beetle larvae on cotton plants not treated with any chemical (CK-Drip irrigation and CK-Foliar spray) and treated with sulfoxaflor through drip irrigation (S-Drip irrigation) and foliar spray (S-Foliar spray) in 2016 (**a**) and 2017 (**b**). All data are expressed as mean ± S.E. (n = 3). Values within the same time period labeled with different letters are significantly different based on Tukey’s HSD test at *p* < 0.05.

**Figure 5 insects-10-00345-f005:**
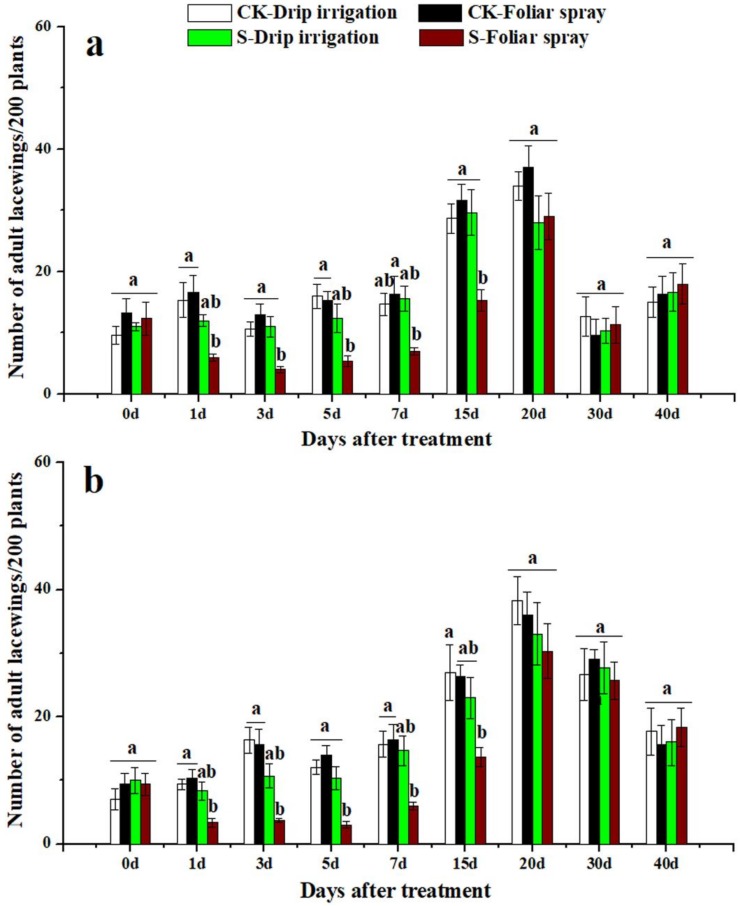
The numbers of adult lacewings on cotton plants not treated with any chemical (CK-Drip irrigation and CK-Foliar spray) and treated with sulfoxaflor through drip irrigation (S-Drip irrigation) and foliar spray (S-Foliar spray) in 2016 (**a**) and 2017 (**b**). All data are expressed as mean ± S.E. (n = 3). Values within the same time period labeled with different letters are significantly different based on Tukey’s HSD test at *p* < 0.05.

**Figure 6 insects-10-00345-f006:**
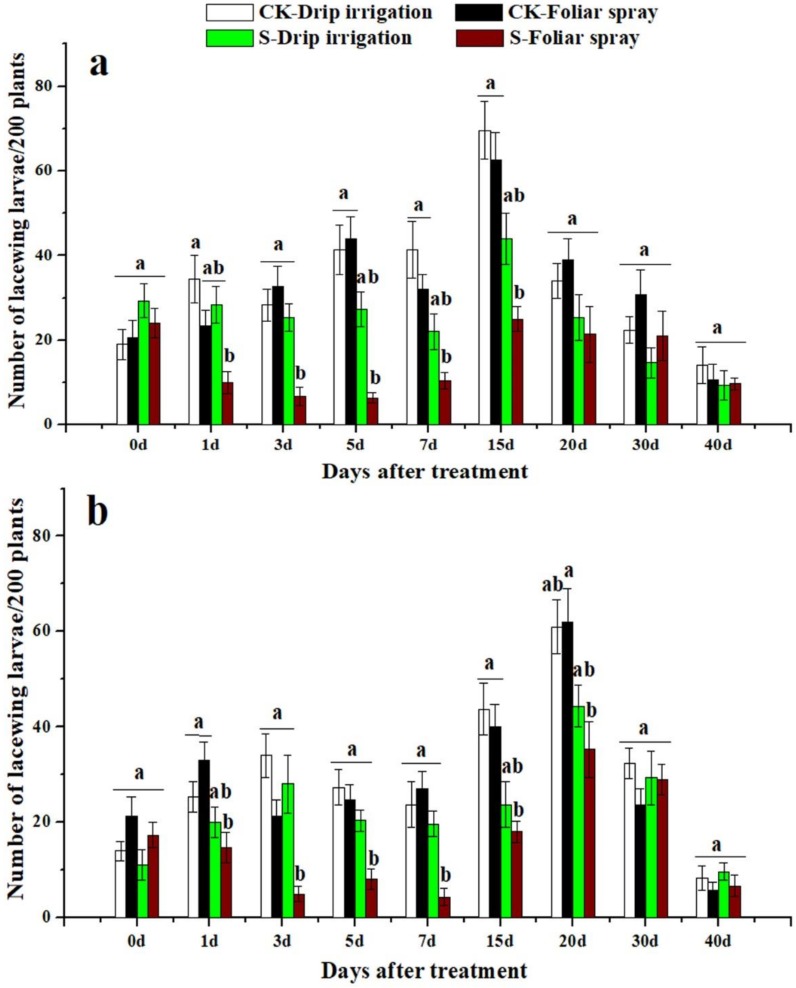
The numbers of lacewing larvae on cotton plants not treated with any chemical (CK-Drip irrigation and CK-Foliar spray) and treated with sulfoxaflor through drip irrigation (S-Drip irrigation) and foliar spray (S-Foliar spray) in 2016 (**a**) and 2017 (**b**). All data are expressed as mean ± S.E. (n = 3). Values within the same time period labeled with different letters are significantly different based on Tukey’s HSD test at *p* < 0.05.

**Figure 7 insects-10-00345-f007:**
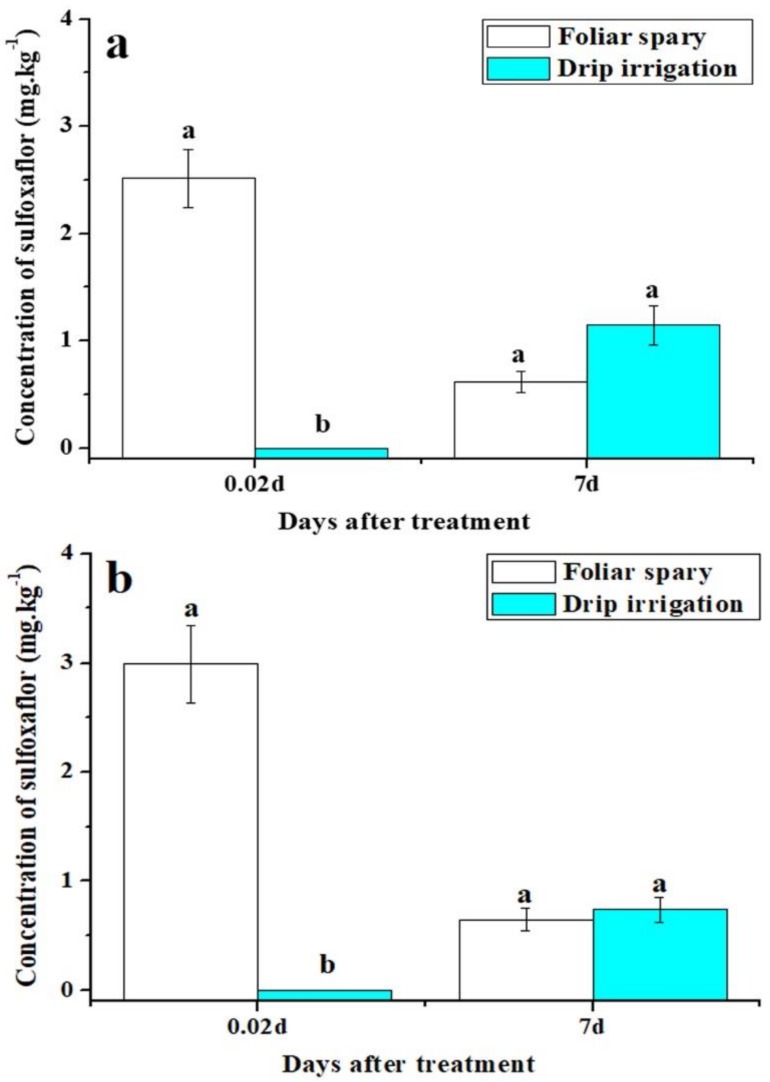
The concentrations of sulfoxaflor in aphids collected from plants applied through drip irrigation and foliar spray in 2016 (**a**) and 2017 (**b**). All data are expressed as mean ± S.E. (n = 3). Different letters above bars within the same time period are significantly different based on Tukey’s HSD test at *p* < 0.05.

**Table 1 insects-10-00345-t001:** Timeline of field experiments conducted in 2016 and 2017.

Dates	Experiments
4 May 2016	Sow seeds
3 June 2016	Counting the number of cotton aphids and two kinds of natural enemies before sulfoxaflor application
3 June 2016	Applying sulfoxaflor through drip irrigation or foliar spray
3 June 2016–13 July 2016	Counting the number of cotton aphids and natural enemies, and collecting cotton root, stem and leaves
24 April 2017	Sow seeds
21 June 2017	Counting the number of cotton aphids and natural enemies before sulfoxaflor application
21 June 2017	Applying sulfoxaflor through drip irrigation or foliar spray
21 June 2017–31 July 2017	Counting the number of cotton aphids and natural enemies, collecting cotton root, stem and leaves

**Table 2 insects-10-00345-t002:** Control efficacies of sulfoxaflor to cotton aphid with two application methods in 2016 and 2017 ^1^.

Time (Year)	Application Method	Leaf Position	Corrected Mortality (%)							
			1 day	3 days	5 days	7 days	15 days	20 days	30 days	40 days
2016	Drip	Top	50.76 ± 5.68aB	80.70 ±3.69aB	96.79 ± 0.35aA	99.00 ± 0.20aA	98.36 ± 0.24aA	92.95 ± 1.27aA	87.84 ± 2.01aA	82.19 ± 2.97aA
		Middle	31.16 ± 6.43abB	63.54 ± 2.62bB	88.53 ± 2.93aA	95.70 ± 0.61aA	94.16 ± 1.18aA	86.22 ± 1.49abA	77.63 ± 3.08abA	72.08 ± 2.08aA
		Lower	12.07 ± 5.39bB	41.28 ± 4.74cB	68.27 ± 3.77bA	78.19 ± 3.37bA	83.79 ± 3.34bA	76.38 ± 3.71bA	68.76 ± 4.67bA	57.41 ± 3.31bA
	Foliar spray	Top	76.82 ± 3.46aA	96.66 ± 0.72aA	95.10 ± 0.86aA	89.51 ± 1.46aB	77.85± 4.72aB	53.96 ± 4.94aB	31.83 ± 5.12aB	2.97 ± 6.67aB
		Middle	70.25 ± 3.02abA	81.15 ± 2.73bA	88.04 ± 2.09aA	78.05 ± 2.78abB	70.59 ± 4.48aB	43.40 ± 3.41aB	21.71 ± 6.23aB	2.55 ± 5.75aB
		Lower	57.00 ± 3.38bA	74.14 ± 2.66bA	75.56 ± 4.24bA	66.00 ± 4.57bA	58.73 ± 5.09aB	39.95 ± 3.69aB	17.01 ± 5.44aB	−3.47 ± 5.36aB
2017	Drip	Top	44.41 ± 3.42aB	75.37 ± 3.48aB	92.02 ± 1.30aA	97.18 ± 1.35aA	96.52 ± 1.08aA	94.86 ± 1.41aA	89.58 ± 3.48aA	70.27 ± 2.90aA
		Middle	28.30 ± 3.81abB	59.18 ± 3.85bB	85.24 ± 3.18aA	92.45 ± 1.12aA	92.28 ± 1.19abA	89.21 ± 2.10abA	80.40 ± 4.14abA	67.23 ± 2.44aA
		Lower	13.58 ± 4.25bB	33.75 ± 3.54cB	72.39 ± 3.13bA	81.39 ± 3.67bA	84.91 ± 2.81bA	79.65 ± 3.79bA	70.58 ± 3.74bA	51.78 ± 3.81bA
	Foliar spray	Top	82.53 ± 4.30aA	94.57 ± 1.04aA	93.36 ± 1.82aA	92.74 ± 1.97aA	80.22 ± 2.44aB	66.91 ± 6.43aB	29.43 ± 5.72aB	18.39 ± 3.84aB
		Middle	78.70 ± 2.57abA	84.34 ± 4.89abA	82.23 ± 3.29abA	81.53 ± 4.01abA	73.18± 5.09aB	57.60 ± 4.42aB	24.53 ± 4.20aB	10.57 ± 4.98aB
		Lower	63.76 ± 3.48bA	72.16 ± 2.69bA	69.42 ± 4.40bA	67.61 ± 3.38bA	54.10 ± 5.01bB	41.35 ± 6.60aB	11.49 ± 5.22aB	6.14 ± 5.57aB

^1^ Data are expressed as mean ± S.E. (n = 3). Data analysis was based on individual year. When significant differences occurred, means were separated by Tukey’s HSD test (*p* < 0.05). Different lower letters in the same columns indicate significant differences among three leaf groups based on the same application method. Different capital letters in the same columns indicate that the same group of leaves have significant differences in corrected mortality due to the application methods.

**Table 3 insects-10-00345-t003:** Estimated annual costs (in United States dollar ($)) in control of cotton aphid with sulfoxaflor applied via drip irrigation and foliar spray during cotton production in Xinjiang, China.

Item	Drip Irrigation ^1^	Foliar Spray
Insecticide cost ($ ha^−1^) ^2^	−202.39	−43.48
Labor cost ($ ha^−1^) ^3^	−1.49	−14.88
Machinery and diesel costs ($ ha^−1^) ^4^	0	−59.52
Mechanical damage ($ ha^−1^) ^5^	+89.29	−89.29
Yield difference ^6^	+95.00	−95.00
Accumulative cost ($ ha^−1^) ^7^	−19.59	−302.17

^1^ The plus and minus signs (+ and −) in the drip irrigation and foliar spray columns represent gain and loss in the United States dollar ($), respectively. ^2^ Insecticide cost: drip irrigation used 700 g a.i. ha^−1^, which was $202.39 per hectare, while foliar spray needs to apply 75 g.ai. ha^−1^ twice, equivalent to $43.48. ^3^ Labor cost included the preparation of insecticide solutions and tractor driver’s salary and benefits. ^4^ Machinery and gasoline costs referred to tractor usage and repair and the cost of diesel for the spray. ^5^ Foliar spray resulted in 2-4% loss of cotton plants, which was a minimum loss of $89.29 per hectare, which was converted to $89.29 gain for drip chemigation. ^6^ Drip applied plants had 5% higher yield than foliar applied plants. ^7^ Accumulative cost is the sum of individual column.

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
