# Peer review of "Sulfoxaflor Applied via Drip Irrigation Effectively Controls Cotton Aphid (Aphis gossypii Glover)"

_insects, 2019, doi:10.3390/insects10100345_

Round 1
Reviewer 1 Report
L179: change “rapidly” to “more rapid”
Figure 1: suggestion for visual presenting: the contrast of black and dark blue is too moderate, changing dark blue to a different color would make the graph easier to read. Same comments to the other figures.
The text font and size in some figures are hard to read, especially figure 5.
L462: “aphidophagous” should be non-italic
I’m glad to see the author explained the rationale behind the huge difference between the application rates of drip irrigation and foliar spray. I think this part should be moved to the material and method section, otherwise the readers will get confused and start questioning the experimental design till the discussion part. However, the application rate of sulfoxaflor via drip irrigation was still based on assumptions instead of preliminary study data, I strongly feel that the author should explain more about how the two different rates were set in details, i.e. how did the author decide to use ~10 fold ai/ha in drip irrigation compared to foliar spray.
Overall the study met the major objectives and answered the questions of the efficacy of delivering sulfoxaflor via drip irrigation vs. foliar spray, except that:
As a study of practical application, there is an important piece missing here, which would be the crop growth/yield/quality. If the benefit of delivering pesticide via drip irrigation cannot compensate the yield loss comparing to foliar spray (this could be the case since foliar spray provided more rapid control), it would be much less valuable in practice. Therefore, it is important to show whether the plant growth or yield increased by drip irrigation treatment is comparable or even better than those treated via foliar spray. I don’t know why this parameter was not included in the study, considering that the cotton was actually harvested in the study for both years.
I don’t think the statement that sulfoxaflor applied through drip irrigation was safer than foliar spray to ladybirds and lacewings are convincing enough based on the experimental design and results. The author also fully realized that the density of prey was a significant factor which affected the population of their natural predators. It is legitimate to conclude that field treated by drip irrigation had higher population of ladybirds and lacewings, but not to conclude that drip irrigation treatment is safer to them than foliar treatment. Based on the data in the manuscript, there was no direct evidence shown that foliar spray or aphid treated by foliar spray resulted in higher mortality in ladybirds and lacewings; there was not enough data to support the author’s conclusion.
Author Response
Response to Reviewer 1 Comments
We really appreciate your suggestions and comments about our manuscript. Based on your comments, we have made revision. Here are our responses to your comments.
Point 1: Change “rapidly” to “more rapid”
Response 1: The suggested change has been made.
Point 2: Figure 1: suggestion for visual presenting: the contrast of black and dark blue is too moderate, changing dark blue to a different color would make the graph easier to read. Same comments to the other figures.
Response 2: Figure 1 has been modified based on your suggestion.
Point 3: The text font and size in some figures are hard to read, especially figure 5.
Response 3: The text and fond size have been modified including Figure 5.
Point 4: “aphidophagous” should be non-italic
Response 4: We have revised it in the new manuscript.
Point 5: I’m glad to see the author explained the rationale behind the huge difference between the application rates of drip irrigation and foliar spray. I think this part should be moved to the material and method section, otherwise the readers will get confused and start questioning the experimental design till the discussion part. However, the application rate of sulfoxaflor via drip irrigation was still based on assumptions instead of preliminary study data, I strongly feel that the author should explain more about how the two different rates were set in details, i.e. how did the author decide to use ~10 fold ai/ha in drip irrigation compared to foliar spray.
Response 5: Thank you again for your comments. We have revised it based on your suggestion. The application rate of sulfoxaflor via drip irrigation was based on our preliminary study data and also in reference to published reports on drip application of other pesticides. The preliminary study data showed that sulfoxaflor effectively controlled cotton aphids when its application rate was 700 g a.i. ha-1. As we mentioned in the manuscript, soil could affect availability of sulfoxaflor. Therefore, we chose that application rate to evaluate the efficacy of sulfoxaflor applied via drip irrigation in relation to foliar spray for control of cotton aphid.
Point 6: Overall the study met the major objectives and answered the questions of the efficacy of delivering sulfoxaflor via drip irrigation vs. foliar spray, except that: As a study of practical application, there is an important piece missing here, which would be the crop growth/yield/quality. If the benefit of delivering pesticide via drip irrigation cannot compensate the yield loss comparing to foliar spray (this could be the case since foliar spray provided more rapid control), it would be much less valuable in practice. Therefore, it is important to show whether the plant growth or yield increased by drip irrigation treatment is comparable or even better than those treated via foliar spray. I don’t know why this parameter was not included in the study, considering that the cotton was actually harvested in the study for both years.
Response 6: Thank you again for your good comments. According to the data showed that the yield of cotton treated by drip irrigation at harvest was about 5% higher than those treated via foliar spray. Such information has been added in this revision.
Point 7: I don’t think the statement that sulfoxaflor applied through drip irrigation was safer than foliar spray to ladybirds and lacewings are convincing enough based on the experimental design and results. The author also fully realized that the density of prey was a significant factor which affected the population of their natural predators. It is legitimate to conclude that field treated by drip irrigation had higher population of ladybirds and lacewings, but not to conclude that drip irrigation treatment is safer to them than foliar treatment. Based on the data in the manuscript, there was no direct evidence shown that foliar spray or aphid treated by foliar spray resulted in higher mortality in ladybirds and lacewings; there was not enough data to support the author’s conclusion.
Response 7: As mentioned above, we have added two figures in this revision, appropriate revision has been made.
Reviewer 2 Report
The title of study needs revision. For example:
Effects of sulfoxaflor via drip irrigation in controlling cotton aphid and its selectivity to two predators species
Introduction, objectives, materials and methods looks fine.
Results, discussion and conclusions need improvement.
Reviewer comments and suggestions are embedded in the attached pdf.

Author Response
Response to Reviewer 2 Comments
Thank you so much for reviewing our manuscript. We have revised the manuscript based on your suggestions and comments. Here are our responses to your comments.
Point 1: Were these mortalities corrected by Abbott formula? if so, this need to be explained in the Data Analyses.
Response 1: Yes, you are right, and we have added it in the revised manuscript.
Point 2: Remove bolds
Response 2: The suggested change has been made.
Point 3: Change to Hours and days after treatment. Use different color schemes for Root, Stems, Leaf and Top leaf, Middle Leaf and Lower Leaf.
Response 3: Based on your suggestions, we have made the revision.
Point 4: ladybird beetles and number
Response 4: We have revised it as you suggested.
Point 5: line 272-275 remove these statements and insert it in Introduction section, these are not already stated.
Response 5: Based on your suggestions, we have deleted it in the revised manuscript.
Point 6: Any observations on the larvae of ladybird beetles or adults of lacewings? If so, we need to report. If not, this need to be mentioned here.
Response 6: We appreciate your valuable comments. In this revision, we have added the results of larvae of ladybird beetles and adults of lacewings.
Point 7: Beetles, only selected stages?? However, we have not tested the larvae of ladybird beetles or adults of lacewings.
Response 7: We also tested the number of the larvae of ladybird beetles and adults of lacewings, two additional figures have been added in this revision.

Round 2
Reviewer 1 Report
I have read the author's reply and revisions. The author has answered my questions and concerns, therefore I'm glad to accept the manuscript in present form.